# Utilizing Drone-Based Ground-Penetrating Radar for Crime Investigations in Localizing and Identifying Clandestine Graves

**DOI:** 10.3390/s23167119

**Published:** 2023-08-11

**Authors:** Louise Lijcklama à Nijeholt, Tasha Yara Kronshorst, Kees van Teeffelen, Benjamin van Manen, Roeland Emaus, Jaap Knotter, Abeje Mersha

**Affiliations:** 1Technologies for Criminal Investigations, Saxion University of Applied Sciences, M.H. Tromplaan 28, 7513 AB Enschede, The Netherlands; l.lijcklamaanijeholt@saxion.nl (L.L.à.N.); jaap.knotter@politieacademie.nl (J.K.); 2Unmanned Robotic Systems, Saxion University of Applied Sciences, Ariënsplein 1, 7511 JX Enschede, The Netherlands; k.j.vanteeffelen@saxion.nl (K.v.T.); b.r.vanmanen@saxion.nl (B.v.M.); a.y.mersha@saxion.nl (A.M.); 3Business, Building & Technology, Saxion University of Applied Sciences, M.H. Tromplaan 28, 7513 AB Enschede, The Netherlands; r.emaus@saxion.nl; 4Police Academy, Arnhemseweg 348, 7334 AC Apeldoorn, The Netherlands

**Keywords:** clandestine graves, drone-based sensors, ground-penetrating radar, radargram, RTK-GPS, obstacle avoidance

## Abstract

The decomposition of a body is influenced by burial conditions, making it crucial to understand the impact of different conditions for accurate grave detection. Geophysical techniques using drones have gained popularity in locating clandestine graves, offering non-invasive methods for detecting surface and subsurface irregularities. Ground-penetrating radar (GPR) is an effective technology for identifying potential grave locations without disturbance. This research aimed to prototype a drone system integrating GPR to assist in grave localization and to develop software for data management. Initial experiments compared GPR with other technologies, demonstrating its valuable applicability. It is suitable for various decomposition stages and soil types, although certain soil compositions have limitations. The research used the DJI M600 Pro drone and a drone-based GPR system enhanced by the real-time kinematic (RTK) global positioning system (GPS) for precision and autonomy. Tests with simulated graves and cadavers validated the system’s performance, evaluating optimal altitude, speed, and obstacle avoidance techniques. Furthermore, global and local planning algorithms ensured efficient and obstacle-free flight paths. The results highlighted the potential of the drone-based GPR system in locating clandestine graves while minimizing disturbance, contributing to the development of effective tools for forensic investigations and crime scene analysis.

## 1. Introduction

A clandestine grave refers to a burial site intentionally hidden or concealing a crime or illicit activity. These graves serve as the discreet location for disposing human remains, contraband, or animal cadavers. Efforts made to ensure the burial sites remain undetected make it extremely challenging for authorities to uncover them. The discovery and excavation of clandestine graves play a vital role in criminal investigations, providing evidence, closure, and justice to victims and their families [1].

Understanding the decomposition process within these clandestine graves is crucial in these cases as it involves intricate processes like autolysis, putrefaction, and decay. Several factors influence this post-mortem transformation, and burial conditions significantly impact the decomposition process [2]. Therefore, comprehending how different burial conditions affect decomposition is essential, particularly in developing accurate methods for locating these concealed graves.

There are several geophysical techniques that could be used to detect clandestine graves using drones. Drones carrying such sensors are gaining popularity amongst law enforcement agencies [3]. They have extended the traditional ground-view outdoor documentation to large-scale sky-view imaging in a non-destructive way. Drones provide geo-referenced data as they carry a global positioning system (GPS) receiver [4], which is an opportunity to collect and analyze on site. However, evaluating, combining, and applying such sensors to collect and process data over a large area is a challenge. Moreover, collecting, processing, and visualizing data from these systems require expertise in using the hardware and software of the sensors. This makes the on-site localization of clandestine graves complex.

Various techniques can be employed to detect unmarked graves while minimizing disturbance to the potential crime scene. Ideally, non-invasive and non-contact technologies are utilized to prospect for irregularities on the surface or in the subsurface. Remote sensing technologies, such as hyperspectral imaging and infrared (IR), are used to detect surface-level irregularities without physical contact, relying on signatures such as depressions in the landscape, visible soil disturbance, temperature or humidity variations, or differences in vegetation patterns [5]. However, in cases where surface-level signatures are absent, geophysical technologies are employed to detect subsurface irregularities. These geophysical applications measure different physical properties of the earth at a given location, either requiring contact with the surface or not. Both contact and non-contact measurements are considered non-invasive, ensuring that the subsurface remains undisturbed [6].

Initial experiments were conducted to identify which technologies could detect graves effectively from an airborne platform. Alongside GPR, magnetometry, electrical resistivity (ER), and electromagnetic induction (EMI) were assessed. While magnetometry and electrical resistivity showed some potential in detecting graves, they were not further developed at this stage due to expected interference from the drone or impracticality in performing measurements from a drone. EMI demonstrated less interference from the drone system and provided the best overall results in detecting graves at the test site. Despite both GPR and EMI being geophysical technologies, they measure different properties of the earth’s subsurface and have distinct requirements for data recording, processing, and operation under a drone system. GPR shows superior applicability compared to other sensor technologies and exhibits the highest level of logical utilization; however, it is important to ensure that the GPR meets the specific operational requirements for drone-based operation. The GPR must be compatible and capable of functioning effectively while deployed under the drone.

GPR operates by transmitting electromagnetic pulses (50–2000 MHz) into the ground, which reflect upon encountering a change in conditions, specifically the subsurface’s ability to hold an electrical charge [7]. By analyzing the data obtained from these reflections, differences in subsurface characteristics can be identified and analyzed without having a destructive quality to the locality [8].

GPR consists of a transmitter antenna and a receiver antenna, which transmit and detect electromagnetic waves at specific frequencies [9]. A reflected signal occurs when an energy pulse enters a material with different conductive properties than the material it originates from. The intensity/size of the reflection is determined by the contrast of the dielectric constant (a characteristic number representing the quotient of electric displacement and electric field strength) and the electrical conductivity of the two materials. The signal attenuation rate varies greatly and depends on the properties of the material through which the electric pulse travels. The signal partially penetrates the ground and reflects when it encounters an object or boundary. These reflections result in changes in frequency, allowing the detection of subsurface structures. GPR can also detect irregularities in the ground or disturbances caused by activities such as disposing of a body in a grave [10].

These characteristics confirm the applicability of GPR for the detection of human remains throughout various stages of decomposition. The technique involves the examination of disturbed soil and other subsurface anomalies, such as indications of excavation or the presence of objects. As a result, GPR can be effectively utilized in all stages of decomposition and in numerous soil types [11]. A previous study demonstrated the high applicability of GPR in sandy soil and gravel, highlighting its ability to achieve a penetration depth of up to 50 m when utilizing a low-frequency antenna [12]. Considering the diverse soil composition in the Netherlands, including sand, clay, gravel, and others, which are subject to ongoing transformations caused by weather and biological processes, the versatile nature of GPR makes it suitable for the majority of these soil types. Given that most of the forensic research will be focused on the top part of the soil, the loss of signal in deeper contexts will not be an issue.

GPR is also applicable in acidic soil and, to a lesser extent, neutral and alkaline soils. In clay soil, GPR works effectively when the clay content is below 10% to ensure optimal signal penetration, while soils with more than 35% clay limit the effectiveness of GPR. Wet clay soils typically have a penetration depth of less than 1.0 m. Additionally, soils with high concentrations of calcium carbonate weaken the GPR signal, and saline soils have a limited penetration depth of only 25.4 cm [13,14]. While the Netherlands has extensive clay areas, understanding the limitations of GPR is crucial in determining the suitable scenarios where the drone-based GPR can be an asset.

GPR presents a wide range of possibilities, but it does come with its own set of limitations. Police and archaeologists face a significant challenge when using ground-based GPR in expansive areas with rugged terrain and dense vegetation due to its time-consuming nature. To enhance the effectiveness of traditional ground-based GPR methods, it is preferable to conduct searches in open-field environments with short grass and minimal trees and brush. This approach minimizes potential interference caused by complex root systems, which can obscure GPR readings and compromise the accuracy of results [15]. Although ground-based GPR was not utilized in this particular research, future studies should be conducted to compare data obtained from drone-based GPR and ground-based GPR methods.

The integration of GPR with drones offers significant advantages in terms of time efficiency and expands the possibilities of data collection in large and previously inaccessible areas. Drone-based GPR is not designed as a technological improvement in terms of sensor capabilities but as a significant impact on usability and practicality compared to traditional ground-based GPR methods. By combining GPR with drones, data acquisition can be completed in a fraction of the time required by traditional ground-based surveys.

Typically, the traditional approach for deploying GPR entails either manually transporting the equipment across the terrain by an investigator or mounting it onto a vehicle alongside a GPS antenna. This approach finds widespread application in domains such as archaeology, forensics, and geological investigations. Both the traditional approach and the utilization of drone-based GPR yield two-dimensional data, which can subsequently be processed into three-dimensional representations. These representations facilitate comprehensive analyses aimed at identifying subsurface structures with pinpoint accuracy [16].

While drones equipped with GPR have recently emerged in forensic applications, drones integrated with geophysical sensors have already found practical utilization. These include the deployment of surveillance drones, in conjunction with a ground control system, at airports, as well as the utilization of aerial, marine, or robotic drones in search operations for missing individuals [17].

For the practical implementation and consistency of the data acquisition of the drone-based GPR, it is important to improve the autonomy of drone control and data collection, reducing the need for direct involvement from the operator or forensic expert in piloting the drone and sensor. In the past five years, research activity in autonomous drone navigation has increased significantly, with a focus on achieving near-complete autonomy [18]. It is important to balance between the level of autonomy and the desired control to account for operator control and cognitive limitations. It is crucial for operators to understand this balance and consider the appropriate degree of autonomy that aligns with the needs and capabilities of the provided case [19].

This research aimed at prototyping a (semi) autonomous drone system, comprising hardware and software, to aid in the search for clandestine graves. The drone system integrates the ground-penetrating radar (GPR) and provides real-time data acquisition, processing, and visualization, assisting in identifying potential grave locations. The research objectives included exploring the application of drone-based sensors, prototyping a modular drone system, developing software for data management and localization, and assessing the feasibility of the developed system. Furthermore, the project aimed at enhancing the autonomy of drone and data collection while minimizing the need for the forensic expert to become an expert pilot. As such, the expert focuses on identifying hotspots.

The principal objective of this research is to establish a functional prototype of the system. Via its implementation, preliminary results will be generated, enabling an assessment of the system’s performance and identifying the subsequent measures required to enhance its reliability.

By successfully developing and deploying the prototype, an initial evaluation of the system’s capabilities and limitations can be conducted. These preliminary results will serve as a foundation for gauging the effectiveness and efficiency of the system in fulfilling its intended purpose. The findings will guide future steps and dictate the necessary modifications and refinements to bolster the system’s reliability.

This article further explains the methodology (Section 2) employed for developing the drone prototype, along with the initial tests conducted using the prototype. The results (Section 3) will display the completed prototype and the preliminary results, highlighting the establishment of a research facility for collecting initial data and testing obstacle avoidance capabilities. Subsequently, the findings will be critically examined (Section 4 and Section 5) by assessing their advantages and disadvantages and determining the future steps to be taken.

## 2. Materials and Methods

### 2.1. DJI M600 Pro

This research used the DJI M600 Pro drone (SZ DJI Technology Co., Ltd., Shenzhen, China), as depicted in Figure 1, to mount the GPR for localizing the clandestine graves. The core aim is for it to be an airborne mobile system and able to carry components necessary for data collection without disturbing the suspected area. The reason is to provide real-time processing and communication while scanning the area.

The DJI M600 was chosen for its integrated, simple, and effective design, allowing for quick setup. It requires minimum maintenance and is known for its reliable performance over extended periods. With a maximum payload capacity of 6 kg, the M600 is compatible with the CBD GPR (Radarteam Sweden AB, Boden, Sweden).

The M600 features smart and safe flight capabilities. Intelligent electronic speed controllers (ESCs) ensure accuracy, safety, and efficiency, while the A3 flight controller automatically adjusts flight parameters based on the attached payload. The A3 can be upgraded to the A3 Pro, which incorporates redundancy and diagnostic algorithms, providing increased reliability. Additionally, the A3 can incorporate the D-RTK GNSS system for centimeter-level accuracy. Furthermore, the M600 boasts extended flight time and long-range transmission capabilities. It offers a range of up to 5 km in unobstructed areas [20].

### 2.2. Drone CBD GPR

The drone CBD GPR, as shown in Figure 2, is a wireless 2-channel GPR kit designed for use with the DJI M600 pro drone. The complete GPR kit includes post-processing software specifically adapted for the drone. The GPR data acquisition system consists of various components designed to facilitate efficient data acquisition and processing. These components include an integrated 2-channel GPR capable of simultaneously displaying shallow and deep targets, operating via Wi-Fi. An on-board computer also incorporates Cobra CBD Antennas, which comprise three multi-frequency antennas (200/400/800 MHz) operating within the 50–1400 MHz band. To ensure uninterrupted operation, on-board computer features an integrated 73 Wh Li-ion battery with a duration of 8 h. Data acquisition is facilitated by the Cobra DAQ software, while post-processing tasks are supported by the Prism 2 software with a 3D export module. Additionally, an external SW-encoder/connector with a resolution of 10 mm is provided. Power and charging needs are met by an external power/charger with Ethernet RJ/45 connectors, along with an external Mascot 2241 battery charger [21].

### 2.3. Enhancing Precision with RTK-GPS

The prototype was further equipped with real-time kinematic (RTK) GPS positioning. This is a satellite navigation technique used to improve the precision of position data down to a centimeter. To realize RTK location precision, a base station is required to be connected to a drone. The base station is fixed on one point, while the drone moves freely. The rover receives corrections from the base station, which then enables centimeter precision [22]. Figure 3 shows the RTK base station and one rover antenna.

In addition to GPR and RTK-GPS, data were also collected using a downward-facing camera and a laser range finder that precisely measured the altitude of the drone. The SF11/C laser is capable of distance measurements of solid surfaces up to an altitude of 120 m and of water up to 40 m [23]. All information was visualized on an iPad using a web interface.

### 2.4. Test Methodology

Over a two-year period, from 2021 to 2023, several tests were performed at the research facility using two drone models, the M600 pro drone and the Acecore Zoe, both depicted in Figure 4. The research facility located in Bentelo conducted 12 tests, which included GPR and obstacle avoidance testing. Additionally, 12 more tests were conducted at Space53, a research facility dedicated to path planning, that solely focused on obstacle avoidance. While a few GPR tests were performed at Space53, these did not involve the use of cadavers but were conducted using the burial of objects. These tests served as input for the subsequent tests in Bentelo and were therefore not further included in this research.

During the initial tests conducted at Bentelo, various combinations of drone altitude and speed were evaluated. These tests began at an altitude of 5 m and a longitudinal speed of 2 m per second, with the altitude being decreased by 1 m and the speed adjusted to 1 m per second. The final round of tests concluded that the optimal altitude and speed for achieving the clearest results were 2 m and 3 m per second, respectively.

Several factors can compromise investigations, resulting in anomalies or false positives. Moreover, when drones fly too close to the surface, forensic or trace evidence can be jeopardized. The primary objective of this study was to determine the optimal flying height and speed for effectively detecting disturbances at the Bentelo research facility. However, maintaining a flight height of 2 m poses a risk of disturbing potential surface evidence. Therefore, additional research is necessary to enhance drone-based GPR best practices. The deployment of drone-based GPR can be tailored to different case types, with some cases requiring higher flight altitudes and others necessitating lower ones in cases with little to no surface evidence. The flight height should be adjusted based on the specific requirements of each case. Furthermore, further investigation is needed to assess if there is any potential data loss when operating at higher flight altitudes.

In this research, an additional requirement has been incorporated into the planning process, namely, the avoidance of obstacles. This is an important functionality to increase the autonomy of the drone as well as expanding the drone’s deploy ability in various areas. When an obstacle is detected, the global path must be promptly redirected in the most efficient manner possible. Unlike global planning, local planning is constantly updated based on real-time sensor data. The algorithm responsible for this task operates on the drone’s on-board computer and communicates with the drone’s flight controller to issue instructions during the flight.

### 2.5. Global Planning

A common method for charting a route across a geographical region involves employing a global planner that makes use of satellite imagery, as shown in Figure 5, the QGroundControl program serves as the tool for carrying out the global planning operations. This planner can autonomously determine a path by analyzing the designated area. The width of the flight path and the altitude can be customized to accommodate the specific sensor requirements of the drone. Once the global path is calculated, it can be uploaded to the drone’s on-board computer for reference during the flight.

### 2.6. Local Planning

After loading the global plan onto the drone’s on-board computer, the local planning phase begins. This involves computing commands to guide the drone along the designated path while avoiding obstacles. Preliminary tests have utilized stereo vision cameras for obstacle detection, employing two side-by-side cameras to provide depth perception. The stereo vision was replaced by 3D LiDAR at the later stage. The primary role of local planning is to guide the drone’s movements and ensure its adherence to the global plan with precision as much as possible while avoiding obstacles.

#### 2.6.1. Obstacle Avoidance

The drone’s advanced technology enables its automated takeoff from the docking station. In Figure 6, the top right corner shows a 3D map of the drone’s surroundings created by the 360 LiDAR system (SF11/C microlidar, LightWare LiDAR, Boulder, CO, USA), while the bottom right displays a live view from the on-board camera. This map, including detailed obstacle information, is utilized for path planning. The drone’s advanced RTK-GPS system ensures a precise and autonomous landing in the docking station, guaranteeing the safe and accurate return of the drone.

#### 2.6.2. LiDAR

Light detection and ranging (LiDAR) uses a pulsed laser to measure distances to the surface. It can be used for terrain following, terrain holding (i.e., precision hovering for photography), improved landing behavior (range aid), warning of regulatory height limits, and collision prevention.

For obstacle avoidance research “Puck LITE” was used. The Puck LITE LiDAR from Velodyne is a lighter version of the VLP-16 Puck, designed to meet the weight requirements of certain applications without compromising performance (VLP-16 Puck Lite, Götting KG, Lehrte, Germany). With a 360° surround view, it captures real-time 3D LiDAR data, including distance and calibrated reflectivity measurements. The sensor offers an unprecedented field of view and point density, with a range of 100 m and dual return mode for enhanced detail in the 3D image. Its compact footprint, low power consumption, and high resolution make it well suited for drone and mobile applications in mapping, imaging, inspection, and navigation. Puck LITE supports 16 channels and generates approximately 300,000 points per second from a 360° horizontal field of view and a 30° vertical field of view (“Velodyne PUCK Lite-Light 3D LiDAR Sensor in Real-Time—ClearPath”). It operates reliably in various environmental conditions, with no visible rotating parts and a wide temperature range. The sensor’s specifications include a measurement range of 100 m, ±3 cm range accuracy, a 360° horizontal field of view, and an operating voltage of 9V to 18 V. It weighs around 590 g and is IP67-rated for environmental protection. Puck LITE connects via a 100 Mbps Ethernet connection, with UDP packets containing various data points such as time of flight distance measurement, calibrated reflectivity measurement, rotation angles, and synchronized time stamps. It also supports GPS integration [24,25].

## 3. Results

### 3.1. Analysis of Requirements

The key aspects of the prototyping process focused on mobility, real-time communication, data processing capabilities, computational power for AI processing, and compactness with the drone. Figure 7 depicts the prototyped drone. The communication between the drone and remote controller was not dependent on the on-board computer and therefore was not included in the requirements. The on-board computer utilized its own processor separate from the drone’s processor. The requirement was to enable real-time access to the on-board computer with a Wi-Fi modem. An operator responsible for controlling the communication unit and a pilot for flying the drone were necessary, ensuring that the pilot’s ability to fly the drone was not disrupted by the communication unit. However, in the case of a single-person data collection scenario, the DJI autopilot application could be used to plan the drone’s route.

The desired range for communication is up to 500 m, allowing for coverage over a significant area. Furthermore, the distance between the “Base station” and the “On-board computer” can extend up to 40 km thanks to the utilization of low-frequency communication facilitated by Lora technology.

The pivotal component that facilitates the flow of information is the “On-board computer” block. This block serves as a hub, connecting all other blocks, sensors, and components. In consideration of security concerns, an on-board computer is incorporated to ensure data processing and the storage of the collected numerical data from the sensors. Additionally, it can possess AI processing capabilities. As depicted in Figure 8, wireless communication between the “On-board computer” and the “Communication unit” is required, necessitating an access point device that allows the “Communication unit” to establish a connection and provide real-time communication with the on-board computer.

The “Communication unit” block assumes the role of controlling, managing, and operating the “On-board computer” and can be regarded as a “remote control” for the on-board computer. It is intended to be utilized by the operator, typically an investigator. The requirement is to establish a means of accessing the information collected and processed by the “On-board computer”. This access is facilitated by a graphical user interface (GUI), enabling the operator to visually inspect the data and interact with the on-board computer using buttons and input fields. This ensures real-time user interaction and inspection with the on-board computer.

The “Communication unit” necessitates wireless communication with the “On-board computer” while maintaining mobility and enabling user interaction and data visualization. The evaluation process focused on the capability to provide both Wi-Fi signals at 2.4 GHz and 5 GHz frequencies. Additionally, the inclusion of a touchpad was considered advantageous as it eliminates the need to carry a separate mouse or keyboard, enhancing mobility. Each concept presented incorporates Wi-Fi and a touchpad.

### 3.2. Functional Design Integration

The integration of the on-board computer for drones involves several critical factors: security, mobility, ease of use, and reliability. In terms of security, the on-board computer must ensure that confidential information is only accessible to authorized personnel, especially since it will be used by the law enforcement. Additionally, data processing and AI analysis should take place directly on the drone, reducing the reliance on a continuous connection with the communication unit. Mobility is essential, requiring the on-board computer to be mountable and meet weight, size, and power limitations to accommodate drone operations. Ease of use is crucial to enable police officers without technical backgrounds to control the on-board computer. The interaction and interpretation of data should be presented in an intuitive and easily understandable manner. Lastly, reliability is vital to ensure the on-board computer consistently performs its tasks. This can be achieved by restricting the number of operators connected to it, allowing only one communication unit to be connected at a time. Priority should be given to the first unit attempting the connection, minimizing the risk of sending conflicting commands or incompatible tasks.

### 3.3. Technical Design

The Cobra Wireless GPR sensor can communicate with the Prism software 2.0, which is used for real-time data collection and post-processing. However, the Prism software 2.0 does not provide access to the raw data collected in real time, which limits real-time AI processing. To overcome this limitation, a custom script was implemented in the back-end server. This script enables communication with the GPR sensor and provides real-time access to the raw data. In the software interface shown in Figure 9, there is a radargram on the left-hand side, generated from the raw data of the GPR sensor. This required modifications to the Obspy library to ensure compatibility with the Prism software for post-processing. On the right-hand side, configuration depth (time) cut-off and soil property options for scanning are available, and a “Start” button initiates the scanning and real-time plotting of GPR data. The “Stop button” prompts the user to save the collected data. Currently, the GPR is used exclusively on the on-board computer for real-time data interpretation and collection.

### 3.4. Bentelo Research Facility

For the experimental execution of the project, various facilities and tools were developed for sample and data collection. These included simulated graves (Bentelo research facility), a drone platform, a sensor package, and accompanying manuals and protocols. *Sus scrofa domesticus* was used as a substitute for human cadavers due to legal restrictions. Scientific evidence supports the similarity in decomposition patterns between *S. domesticus* and human cadavers, attributed to their comparable muscle-to-fat ratio, skin characteristics, and body hair [26]. Numerous studies have investigated the disadvantages of *S. domesticus* as an analogue for human remains [27,28,29,30,31]. While analogues, particularly of large-bodied species, serve well in “proof-of-concept” studies, their validity as substitutes for humans varies depending on the specific traits being examined. In the case of studying cadavers buried underground, the initial validation of forensic methods can be efficiently performed using pig cadavers, considering the traits that need to be evaluated [26].

It is important to note that the dissimilar decomposition patterns between a model animal and humans do not render the model completely useless. Its applicability depends on the specific research question being addressed. Given that the purpose of this research facility is to serve as a test site for preliminary results of the prototype being developed, *S. domesticus* served as a suitable analogue in this context [26,27,28].

The Bentelo research facility site was established in November 2018 to simulate clandestine graves with the use of *S. domesticus* cadavers. The total area is 3.4 hectares in size and is in an area surrounded by meadows. The soil composition of the test site consists of sand.

When creating the research facility, the contours of the grave did not precisely match the reality that is seen in practical cases. However, this slight deviation from accuracy did not hinder the overall goal of the research. The objective was to investigate the feasibility of identifying disturbances using a drone-based GPR.

The research site initially contained 12 graves containing *S. domesticus* cadavers and four reference graves. A reference grave refers to a site that does not contain any *S. domesticus* cadavers but rather serves as a benchmark or comparison point for evaluating other graves. These reference graves display the same soil disturbances that later can be used for comparative analysis when analyzing the graves containing *S. domesticus* cadavers.

Two reference graves were strategically placed near the research facility, both equipped with data loggers and left undisturbed. Meanwhile, the remaining two graves were located within the research facility and were similarly equipped with data loggers. However, to simulate actual graves, they were initially dug up and then closed again. The cadavers had an average length of 87 cm with a height of 50 cm and were buried at various depths at 30, 60, 90, and 120 cm. Graves were placed approximately 3 m apart to avoid cross-contamination [32]. The alignment of the graves is visually represented in Figure 10A,B.

The first twelve cadavers were obtained from the slaughterhouse and were approximately 10 to 12 weeks old when slaughtered. The *S. domesticus* were euthanized using an electric shock, followed by bleeding out by a throat incision. All intestines were left intact within the bodies.

Previous research results highlight uncertainties in understanding the dependencies of external factors and the absence of a clear bloat phase. Based on these findings, the research expanded to include larger cadavers with intact organs and blood by alternative slaughtering methods to ensure cadaver integrity and eliminating factors to create a concise depiction of decomposition. In October 2021, four new *S. domesticus* specimens were humanely euthanized using lethal doses to ensure their preservation with intact blood and organs. Of these cadavers, two were positioned with a tube connecting their stomachs to the surface, enabling the documentation of the stages of decomposition in various ways. This allowed for clearer documentation purposes and allowed for more credible sensor data. The exact arrangement of the graves is shown below in Figure 11. Table 1 shows the dimensions of the cadavers present within each grave. 

#### 3.4.1. Ground-Penetrating Radar Analysis Research Facility Bentelo

The M600 drone was employed to survey the cadaver research facility, cruising steadily at a velocity of 3 m per second while maintaining a consistent altitude of 2 m. Several tests were conducted to determine the optimal speed and altitude, but it was found that 2 m provided the most accurate data. Figure 12 illustrates an unmanned autonomous flight conducted at a 5 m altitude using the M600 drone. The results indicate that autonomous flights generate smoother trajectories in comparison to manually flown paths, as depicted in Figure 13. Nonetheless, it is important to note that the M600 drone is incapable of conducting autonomous flights below 5 m due to inherent restrictions.

The drone system successfully generated real-time data points during its scanning of the research facility. This valuable data can be accessed on any system that is connected to the drone’s Wi-Fi center and is logged on to the “hidden graves” system that has been specifically designed for this purpose. Due to the on-board RTK-GPS, the developed system has the capability to determine the precise location of each GPR measurement (trace) and plot them on a map, either in a separate GIS or in the Prism2 software 2.0. 

In addition to Prism2’s data analysis capabilities, end users can perform real-time analyses of crime scene data by accessing hyperbole analysis from any device connected to the drone’s Wi-Fi center. This feature enables faster and more efficient crime scene investigations.

#### 3.4.2. RTK-GPS Integration

The integration of RTK-GPS on the M600 with GPR was initially evaluated above the cadaver research facility. The GPS logs below showcase the performance of the drone in two different flights—one in black without RTK and the other in red with RTK. With the improved accuracy of position logging, areas of interest can be better located in the field, which are visible as hyperbolas in the radar profile. The data are later analyzed using Prism2, enabling precise and efficient post-flight analysis shown in a GPS map.

Based on the preliminary results, several modifications were implemented to enhance the test platform. The installation of a more sophisticated RTK-GPS system has improved the localization accuracy. Moreover, the stereo vision camera was substituted with a 360° LiDAR, enabling the on-board computer to observe obstacles in the surroundings without being affected by lighting conditions. Finally, all these upgrades were installed on a larger drone with a higher payload capacity.

#### 3.4.3. Signal Analysis

During the flight tests conducted at the research site, two radargrams (Figure 14 and Figure 15) were generated. In the first radargram, the take-off point of the drone is visible at trace 1200, followed by a downward and erratic movement of the first reflection (after the direct airwave) until the first “line” is traversed at trace 2000. The direct airwave can be easily distinguished in these radargrams as the distance between the antenna and the ground during the significant measurements was 2 m. The variation in the first reflection after the direct airwave in the radargram can be interpreted as the drone ascending in altitude while the earth surface beneath the GPR lowers.

Subsequently, three lines were manually flown at a nominal height of 2 m until trace 5570. Beyond this point, the first reflection deviates as the drone concludes its mission and returns to the landing platform.

The irregularities observed in the first reflection between trace 2000 and 5570 can be attributed to surface irregularities and elevation corrections. Additionally, minor adjustments in elevation were made at the start and end of each line, resulting in peaks and artifact hyperbolas visible in the radargram.

During the initial flight, a relatively clear and uniform image of the subsurface (Figure 16) was obtained. The surface of the earth is easily identifiable as the first reflection, and no significant anomalies can be detected beneath it. However, certain graves do produce minor reflections, but these can only be recognized as such by comparing the GPS locations of the graves with the corresponding traces. Without prior knowledge, it is challenging to differentiate these anomalies from naturally occurring irregularities in the field that are unrelated to graves.

The radargram from the second flight exhibits similarities to the first radargram in its overall layout. The take-off of the drone is observed at trace 450, the start of the first line at trace 1500, the completion of the 11th line at trace 6800, and the landing of the drone at trace 8000. However, this flight was manually conducted without the use of automatic grid piloting software (QGroundControl, version 2021), resulting in a less regular flight plan consisting of multiple short lines instead of a few longer lines.

The interpretation of this radargram led to the positive identification of several unmarked graves. Specifically, graves no. 9, 12, and 16 were recognized as clear hyperbolas as depicted in Figure 17. These graves appeared as anomalies at or just below the surface of the earth. No additional anomalies were identified at greater depths beneath these anomalies.

### 3.5. Ground-Penetrating Radar Analysis Partner Case Study

To evaluate the efficacy of drone-based GPR in alternative settings, a partner procured a test facility featuring an archaeological site. With the archeological site having undergone extensive analysis via conventional geophysical methods in the past, it served as a suitable location for comparison with the drone-based GPR. The GPR tests yielded positive results, successfully identifying segments of the archeological site. Figure 18 portrays numerous hyperbolas that correspond to a wall within the archaeological site.

### 3.6. Global Planning

Initially, the global planner integrated into QGroundControl had certain limitations. It lacked the capability to identify no-fly zones, which led to flight plans extending beyond the designated area of interest. However, an enhanced global planner has been developed to effectively map out these no-fly zones. This improvement ensures that the drone’s flight plan remains confined within the intended area, making it a viable and dependable option.

### 3.7. Local Planning

Initial laboratory tests have successfully demonstrated the working principle of the obstacle avoidance system. A 3D map is created and continually updated, allowing the on-board computer to calculate the most efficient path for the drone while factoring in obstacles, drone dimensions, and flight limitations such as speed and maneuverability. The drone’s position is accurately tracked using an OptiTrack system that utilizes multiple cameras. This method replaces GPS, which is not available in indoor environments.

## 4. Discussion

### 4.1. Ground-Penetrating Radar

Although the GPR antenna was specifically designed to be mounted on a drone, there was still some development needed, first, to improve precision and, second, to improve operation. One of the goals of the research was to design a complete system that could be operated by a trained but non-specialist user. Therefore, a customized control interface for the GPR system was created so the measurements could be controlled from any mobile device up to 400 m from the drone whilst still providing real-time graphics of the current measurements. The main advantage of this is that users can customize the graphical interface of the on-board computer for any target group. Another advantage is that, in time, additional algorithms can be added into the on-board computer to enhance the interpretations of the data or, even, to automatically detect the graves for the operator via artificial intelligence.

Since the on-board computer is custom designed, it can easily be improved, or secondary systems may be added to upgrade the measurements. To improve the interpretation of the radargram, a laser altimeter was added to the drone. This would track the exact height the drone was flying above the earth surface at every GPR measurement and log it directly to the on-board computer. In terrestrial GPR measurements, the distance from the antenna to the earth surface is fixed because the antenna is typically carried by a cart or sled, thus neatly following the terrain. The inclusion of a drone-mounted antenna introduces two key variables into the radargram. Firstly, it accounts for the variability in the terrain being surveyed. Secondly, it considers the fluctuations in the drone’s altitude during flight. The combination of these variables determines the height above the ground surface.

After determining the optimal parameters for flying the drone with the accompanying antenna setup, multiple scans were conducted on the buried cadavers. The test site posed several challenges, as mentioned earlier: (1) subsurface anomalies could arise from both the cadavers themselves and the soil disturbance caused by their presence; (2) the cadavers were devoid of objects or clothing, which is particularly relevant for the interpretation of the GPR signal; and (3) the graves exhibited varying surface morphology at ground level, characterized by different levels of irregularity. The surface variability especially had a significant impact on the interpretation of the measurements as measurements at the top of the soil often lead to signal artifacts that obscure the underlying signals. Detecting *S. domesticus* proved to be exceptionally challenging, despite the presence of surface artifacts.

Additional evaluations of the results obtained by drone-based GPR in comparison to ground-based GPR surveys of unmarked graves is required. The current drone-based GPR was tested using random search and grid search techniques. The grid search techniques utilized 1 m transects, aligning with the Dutch police’s missing-persons-based search and rescue protocol. The current grid searches were sufficient for the conducted investigation, but there are possibilities to adjust the transect size depending on the target being sought. This may not be feasible with the current system as DJI software makes it challenging to plan flight lines, restricting altitude to above 5 m and preventing close spacing. Nevertheless, all the necessary tools are available with this research to achieve this objective. The ongoing development of proprietary software for flight planning enables the precise design of flight paths according to specific requirements. The drone-based GPR surveys can achieve a positioning accuracy of 1–3 cm with the assistance of RTK-GPS. Whether the drone can accurately follow the planned flight lines depends on several factors, primarily wind, which can divert the drone from its intended path. However, taking all factors into account, we still have approximately 20 cm of leeway compared to the mentioned 25 cm requirement. Therefore, achieving the desired flight line accuracy should be feasible, except during extreme weather conditions.

Due to the irregular flight plan of the second flight, some graves were not traversed by the GPR, resulting in their absence in the radargram. Conversely, one specific grave (no. 12) was inadvertently visited three times during the flight, resulting in three consistent reflections of that grave. This suggests that the system has the capability to detect surface and potentially subsurface anomalies. Therefore, further research and adjustment of automated grid pilot software hold significant potential.

All the graves detected using the drone-based GPR appeared as distinct hyperbolas either at the surface or at shallow depths. This indicates that the detection primarily focuses on the disturbances caused by the excavation and reburial of graves. The presence or absence of anomalies at greater depths, such as buried objects or lower portions of the graves (soil disturbances), remained undetected. However, this does not imply that drone-based GPR cannot detect such deeper anomalies. It is possible that the deeper anomalies may have been masked by the clearer reflections from the upper soil layers. To address this, further research involving graves or buried objects with topsoil homogenized with the surrounding area is recommended.

### 4.2. Research Facility

The current study has used *S. domesticus* as an analogue for humans remains. While these preliminary results have been necessary for initial assessment, it is crucial to emphasize that further testing will be conducted to validate these methods using human cadavers. Future work aims to ensure the applicability and reliability of the techniques in real case scenarios involving human remains.

Furthermore, the graves present at the research facility were specifically dug for research purposes and were not intended to replicate or simulate active cases. The purpose of these preliminary experiments was to assess whether the drone-based GPR could effectively detect disturbances in the ground. The specific focus on grave contours was of secondary importance at this stage. Rather, the project aimed to establish a baseline understanding of the capabilities and limitations of the drone-based platform.

Further research into comparing the drone-based GPR to the conventional ground-based GPR will delve deeper into the detection of grave contours. This will involve a more comprehensive analysis specifically aimed at accurately identifying and mapping the outlines of graves. However, in this initial phase, our priority was to conduct a preliminary assessment to determine if the drone-based platform was even capable of identifying graves.

By prioritizing the broader objective of identifying disturbances and evaluating the drone-based GPR’s effectiveness, the project laid the foundation for future investigations. These subsequent studies will provide a more detailed and thorough examination of grave contour detection and its application in archaeological or forensic contexts.

The initial experiments were crucial for setting the stage and confirming the potential of the drone-based GPR system. By establishing the viability of detecting disturbances from an aerial platform, we paved the way for further advancements and refinement in our research, leading to the more accurate and precise detection of grave contours in subsequent investigations.

### 4.3. Obstacle Avoidance

Several tests have been conducted on the initial versions of global and local planning, and the planners have been fine-tuned to create a cohesive system that encompasses the entire area and avoids obstacles. It is imperative not only to avoid obstacles and reach waypoints but also to precisely follow the designated trajectories to ensure that the sensors beneath the drone effectively cover the area and reach the destination by taking the shortest route while navigating around the obstacles.

In this research, various interdependencies have become known, which primarily affect the obstacle avoidance feature of the on-board computer. The precise position and height estimation of the drone are critical to generate an accurate map of the obstacles that must be avoided. To further enhance accuracy and reliability, it may be feasible to integrate the data from the 360° LiDAR, RTK-GPS, and height measurements.

### 4.4. RTK-GPS

In addition to the vertical accuracy of the measurements, the horizontal accuracy also needs to be improved. The on-board computer is typically delivered with a code-based GPS-tracker, thus providing a planimetric accuracy of up to 10 m. This accuracy is insufficient to pinpoint any subsurface anomalies back in the field once they are detected in the measurements. Therefore, a frequency-based GPS antenna with base-station is included into the on-board computer. With this setup, every GPR-measurement could be recorded within centimeter precision.

The impact of weather conditions on the functionality of RTK-GPS has been noted to significantly affect the quality of the 3D map and obstacle avoidance performance. Additionally, it is evident that various sensors are mutually complementary. However, the utilization of drones entails limitations in terms of space and weight, necessitating the need for informed choices.

## 5. Conclusions

Following the testing of the prototype, preliminary results were achieved pertaining to the visualization of graves at the Bentelo research facility. As the project primarily concentrated on developing a drone-based platform, there is considerable potential for further research to validate the drone-based GPR technology. Future work will aim to conduct further benchmarking and detailed analysis to compare the advantages and limitations of the drone-based GPR to different ground-based GPR platforms.

The system has been proven to work efficiently and with high precision, facilitated by the development of a custom operating system for the antenna. This creates opportunities for the advancement of the system development, such as customizing the interface and operation for specific applications or designing automated object detection algorithms tailored to specific requirements.

Further research determined that the on-board computer is fully deployed and capable of efficiently processing the data collected by the drone. Furthermore, the AI-generated results are presented to the client in real time. Additionally, it has been ascertained that the on-board computer is designed to accommodate other geophysical sensors in the future. This is due to the fact that the on-board computer can collect to the centimeter precision RTK-GPS and LiDAR data for each GPR data sample. Implementing pixel classification on a GPU for shorter prediction time is advised. It would allow the growth of the image resolution while maintaining the prediction speed or faster prediction speed while maintaining the resolution of the image.

It is also recommended to implement object detection of hyperbolas using a GPU for the GPR data. This recommendation is due to the extreme difficulty of open curve recognition even by experts, and, therefore, AI can be a significant advancement.

## Figures and Tables

**Figure 1 sensors-23-07119-f001:**
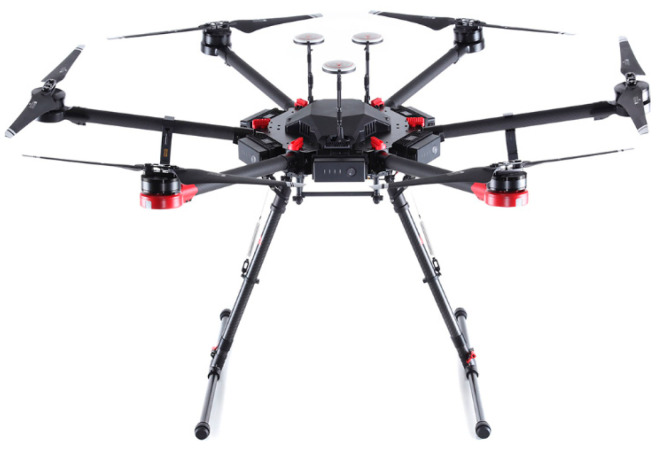
DJI M600 pro [20].

**Figure 2 sensors-23-07119-f002:**
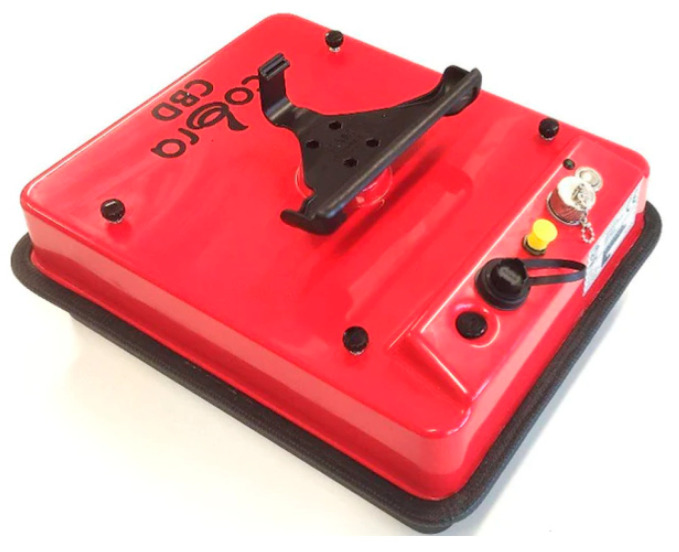
The drone CBD GPR that is in use with the DJI M600 pro [21].

**Figure 3 sensors-23-07119-f003:**
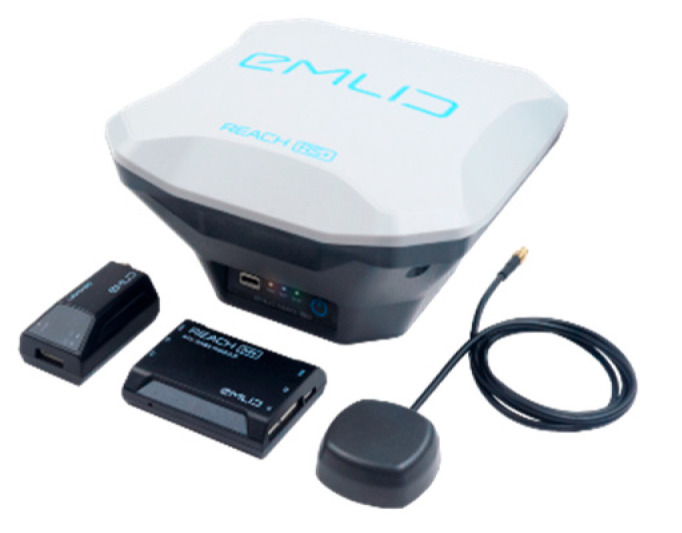
Reach M+ drone mapping kit [22].

**Figure 4 sensors-23-07119-f004:**
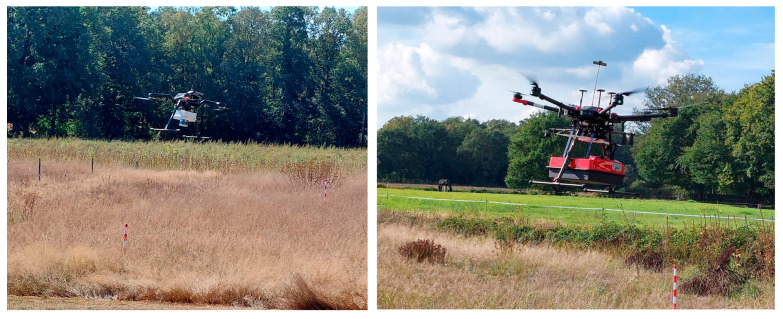
The left side of the figure depicts the Acecore Zoe used for obstacle avoidance. The right side of the figure depicts the DJI M600 combined with the GPR.

**Figure 5 sensors-23-07119-f005:**
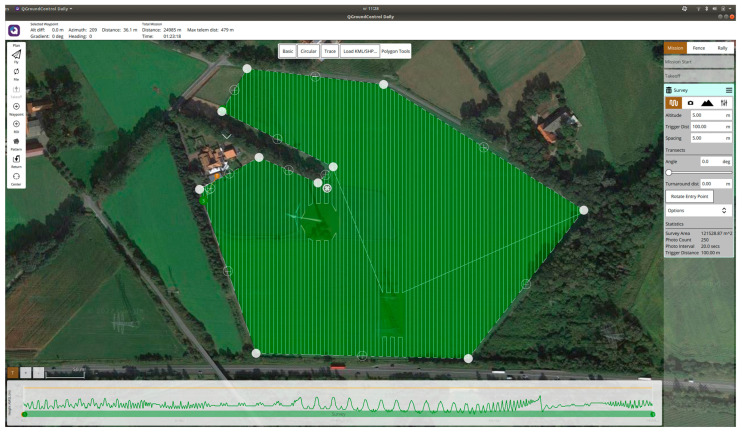
Example of the QGroundControl program that shows the global planning of a flight path that includes a no-fly zone.

**Figure 6 sensors-23-07119-f006:**
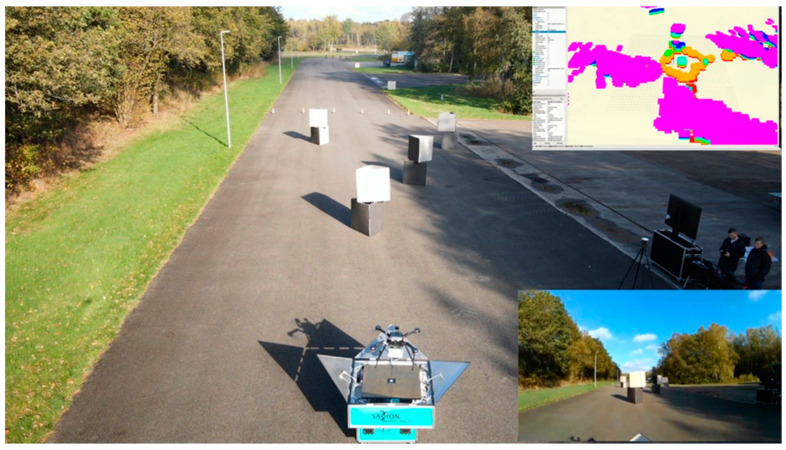
An overall view of the obstacle avoidance testing area at Space 53 depicting the obstacles, the camera viewpoint of the drone (right bottom corner), and the 3D map (right top corner).

**Figure 7 sensors-23-07119-f007:**
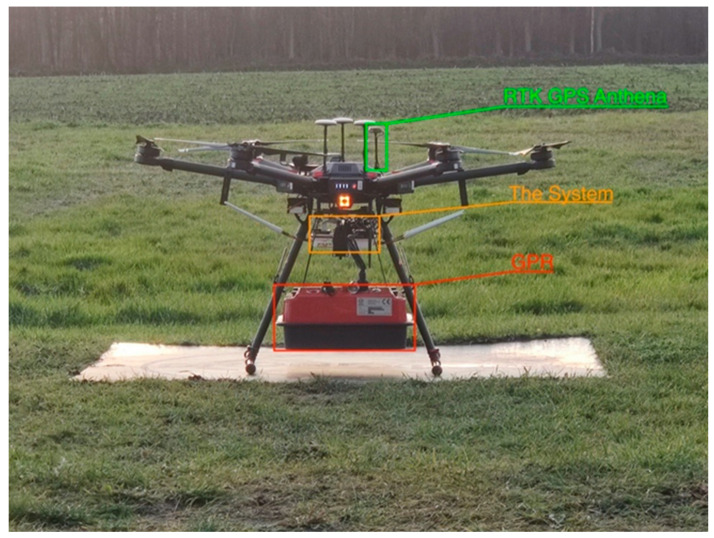
The completed prototyped inspector drone.

**Figure 8 sensors-23-07119-f008:**
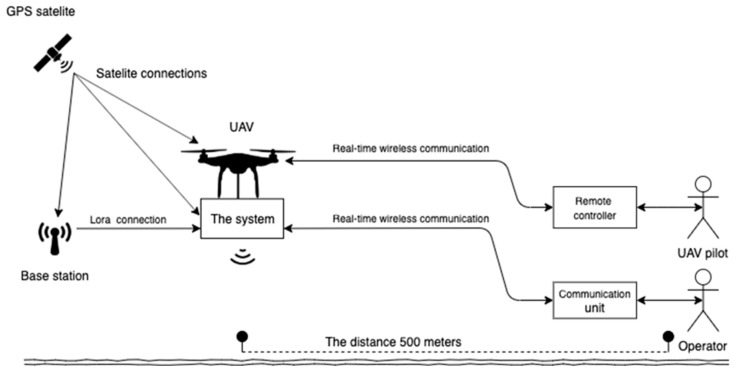
The use case of the on-board computer.

**Figure 9 sensors-23-07119-f009:**
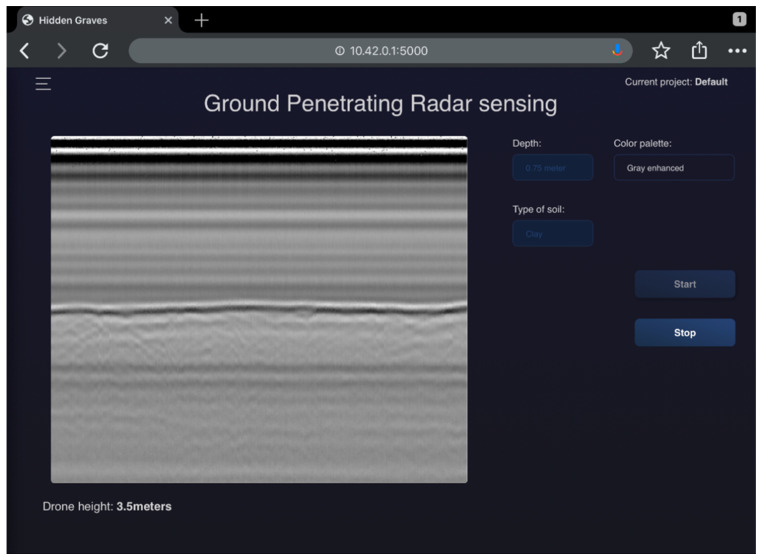
Control experiment: GUI screenshot of a GPR route recorded above Bentelo with grave avoidance.

**Figure 10 sensors-23-07119-f010:**
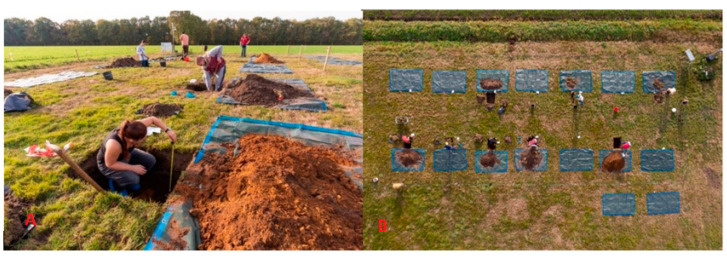
(**A**) The first digging of the 12 graves where the graves are equipped with data loggers, and the soil layer structure is noted. (**B**) An overall view of the research facility depicting the 12 *S. domesticus* graves and four control graves.

**Figure 11 sensors-23-07119-f011:**
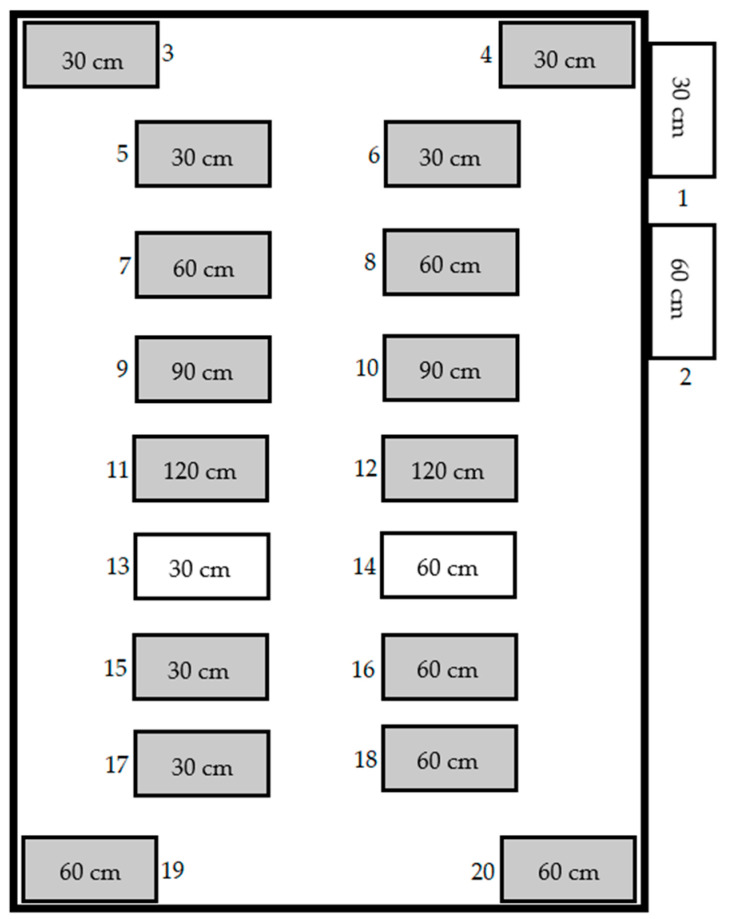
The arrangement and placement of graves at the Bentelo Research Facility.

**Figure 12 sensors-23-07119-f012:**
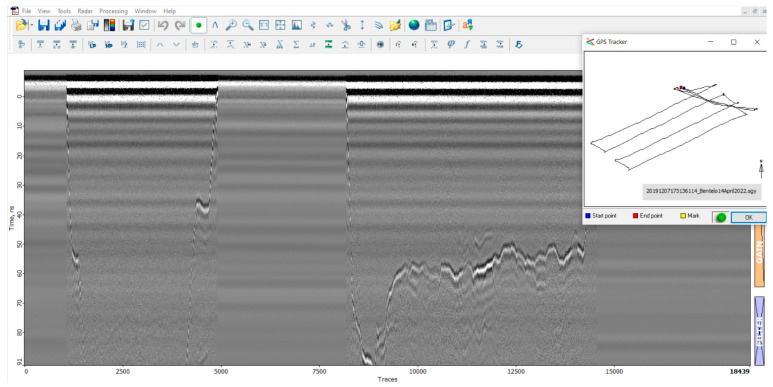
Prism2 screenshot from an autonomous test flown with the M600 Pro in April 2022.

**Figure 13 sensors-23-07119-f013:**
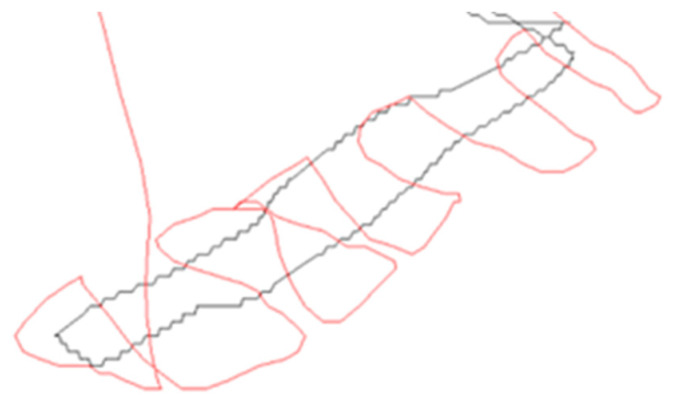
The red line depicted denotes the trajectory flown by the M600, which was integrated with the advanced RTK-GPS system. On the other hand, the black line denotes the route flown without the RTK-GPS system and highlights the lack of centimeter precision.

**Figure 14 sensors-23-07119-f014:**
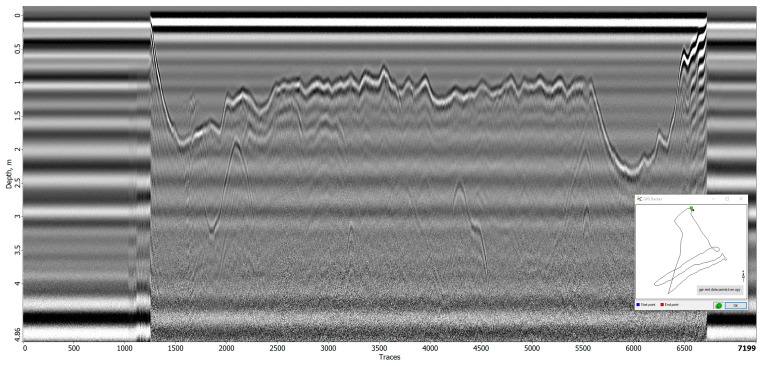
Radargram of the entire first flight.

**Figure 15 sensors-23-07119-f015:**
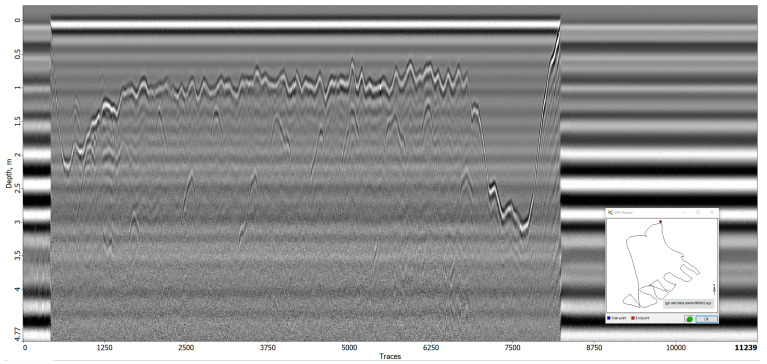
Radargram of the entire second flight.

**Figure 16 sensors-23-07119-f016:**
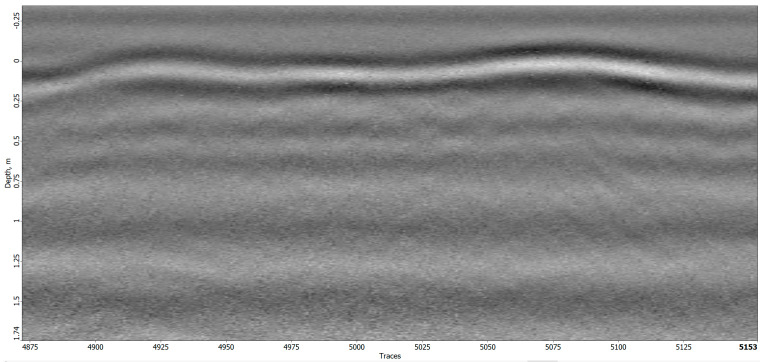
Segment of the B-plot of the first flight, showing only the eastern half of the southern line.

**Figure 17 sensors-23-07119-f017:**
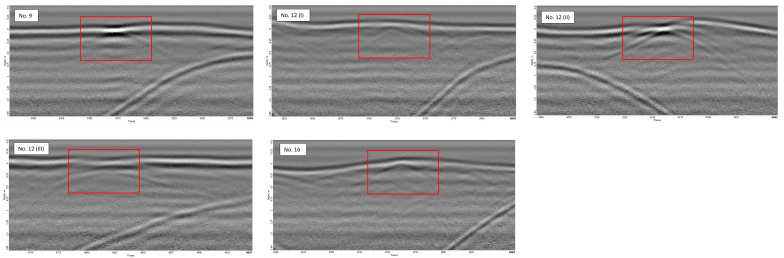
Radargram of the detected graves no. 9, 12, and 16.

**Figure 18 sensors-23-07119-f018:**
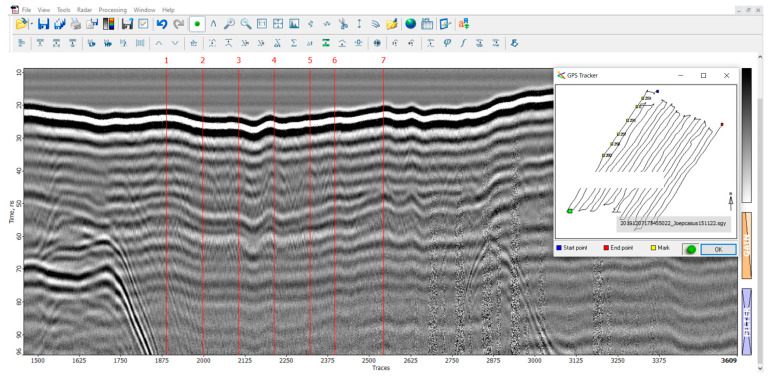
Prism2 screenshot that depicts several hyperboles linked to GPS coordinates (GPS map in the right corner).

**Table 1 sensors-23-07119-t001:** Shows the exact dimensions of the *S. domesticus* cadavers correlating it to their representative grave depth and grave number. Grave number 3, 4, 13, and 14 are reference graves. Cadavers with unknown dimensions are the cadavers added later to the research facility.

Grave Number	Grave Depth (cm)	Cadaver Width (cm)	Cadaver Length (cm)	Cadaver Height (cm)
1	30	-	-	-
2	60	-	-	-
3	30	Unknown	Unknown	Unknown
4	30	Unknown	Unknown	Unknown
5	30	82	15	41
6	30	83	14	50
7	60	96	16	53
8	60	82	15.5	48
9	90	92	17	51
10	90	90	16	48
11	120	83	14.5	49
12	120	87	15	58
13	30	-	-	-
14	60	-	-	-
15	30	84	15	46
16	60	82.5	12.5	46
17	30	89	14	46
18	60	91	15.5	48
19	60	Unknown	Unknown	Unknown
20	60	Unknown	Unknown	Unknown

## Data Availability

The data presented in this study are available on request from the corresponding author.

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
