# Peer review of "Utilizing Drone-Based Ground-Penetrating Radar for Crime Investigations in Localizing and Identifying Clandestine Graves"

_sensors, 2023, doi:10.3390/s23167119_

Round 1
Reviewer 1 Report
This manuscript provide a drone-based GPR system for buried cadavers detections. It is well-written and I have some suggestions:
1. A summary about the aims of this work and the organization of this paper are needed at the end of the Introduction part.
2. The contours of the graves excavated in the experiment were too regular. Are they consistent with the reality
3. The section 2.4 Research Facilities is better to be setted near the experiment results of section 3.4.
4. Can the frequency of the GPR be raised to improve the resolution of results to obtain the information about the shape of the graves?
Minor editing of English language is required such as the title of section 3.4.
Author Response
Dear Reviewer,
Thank you for your thorough review of our paper titled Utilizing drone-based GPR for crime investigations in localizing and identifying clandestine graves. We appreciate your valuable feedback and suggestions for improving the clarity and organization of our work. In response to your comments, we have made the following revisions:
A summary about the aims of this work and the organization of this paper are needed at the end of the Introduction part.
We have addressed this concern by adding a comprehensive summary of the aims and objectives of our study, as well as a clear outline of the paper, at the end of the Introduction section. This addition will provide readers with a better understanding of the goals of our research and the structure of the paper.
The contours of the graves excavated in the experiment were too regular. Are they consistent with the reality?
We acknowledge that the contours of the graves excavated in the experiment appeared to be too regular. However, it is important to note that the research field was primarily used as a control site to investigate disturbances, rather than specifically focusing on the identification of graves. To address this concern, we have extended the results section to include information about the research facility and have provided a separate discussion to explain this further. This clarification will ensure that readers understand the limitations and context of our experimental setup.
The section 2.4 Research Facilities is better to be settled near the experiment results of section 3.4.
We agree with your suggestion and have relocated the section 2.4, "Research Facilities," to a position closer to the experiment results of section 3.4. This reorganization will improve the flow of the paper and enhance the logical progression of the information presented.
Can the frequency of the GPR be raised to improve the resolution of results to obtain the information about the shape of the graves?
In the discussion section, we have addressed this point by explaining that while improving the resolution of the results to obtain information about the shape of the graves was not our primary focus in this study, it is indeed a valid consideration. We have highlighted that further research can explore the possibility of raising the frequency of the Ground Penetrating Radar (GPR) to enhance the resolution and obtain more detailed information about the shape of the graves. This addition acknowledges the potential for future investigations and opens avenues for further exploration in the field.
Furthermore, we have carefully reviewed the entire report and made necessary edits to rectify any grammatical errors and improve the overall language and clarity.
Once again, we would like to express our gratitude for your valuable feedback, which has significantly enhanced the quality of our paper. We believe that the revisions we have made in response to your suggestions have strengthened the paper and improved its readability. We are confident that these changes will address the concerns you raised and provide a more comprehensive understanding of our research.
Thank you for your time and consideration. We look forward to hearing from you soon.
Sincerely,
Louise Lijcklama a Nijeholt
Reviewer 2 Report
Dear Authors,
I enjoyed reading your interesting article. There is a lot of food for thought, and, as it is unique, it is a work that should be published. Nevertheless, there are some points that perhaps you should clarify or better express for completeness of scientific information:
1. The characteristics of the GPR used should be better explained. It is a multifrequency GPR but has a center frequency of 500MHz. In addition, these are unshielded antennas that not only experience noise from below the antennas but also and especially from the surrounding area, aerial in particular (the drone itself for example). Next, the airwave/groundwave dynamics is not mentioned, which is particularly delicate in such a context where you have a layer of air crossed by the reflected wave. These are important aspects to understand and write about because they open the dispute to commentary and conjecture about the actual feasibility and usefulness of such an approach at the crime scene;
2. Not only impediments can obstacles at the crime scene creating anomalies or false positives, but also the presence of forensic evidence or trace evidence on the surface could be compromised by flying too low due to the drone blades creating a wind effect. This seems unimportant, but it is critical from the perspective of proper forensic survey and crime scene contamination;
3. The issue of pigs being used as human body mimics is a much debated topic and not everyone agrees and this point should be better explored especially from a crime scene use perspective (see for example, i. https://academic.oup.com/fsr/article/5/4/249/6794622; ii. https://l.facebook.com/l.php?u=https%3A%2F%2Fwww.nytimes.com%2F2016%2F06%2F14%2Fscience%2Fforensic-science-body-farm.html%3Ffbclid%3DIwAR3PEAtY_32UNS0qbh4AtM3dsNknmoD9oVm6rx8F7urm_hbBSx9IjdYBtJM&h=AT1cVvnwPAKCBX_pGi0HfpnjaMVAfSZlpWdAGlq8tDEVSndE3XVNOLyMyd9pHoenpWcDxVJEd5GDKg34XdSaxClGnUgdHiRCPpRvZR6GeMxsl5hFkZiqwL6_rg2QcumfjaDC&__tn__=-UK-R&c[0]=AT2PgGdyrdcurkHx95avizDwi3lUJ_OUvkIkG_4ogRpbE3zGgW8siR9LzOgV1M_PCAT9qcVryBmxCv6IujlHqwwkAeIaCWteQtUmjUes0uajgQDwDdioV0fh1Q0LZTv711dGCLnrYR2WFGU4pU-YuoTKTZ6-; iii. https://www.mdpi.com/2673-6756/2/4/46);
4. The introduction should also consider other drone+gpr approaches where there is a "traditional" approach of evaluative use by drone and analysis of gpr data from the ground, such as this one for example: https://www.mdpi.com/2504-446X/4/2/15
Despite these critical issues, I believe the paper should be published with particular relevance after a major revision.
Author Response
Dear Reviewer,
Thank you for your valuable feedback and suggestions regarding our paper titled Utilizing drone-based GPR for crime investigations in localizing and identifying clandestine graves We appreciate your thorough review and have made the following revisions in response to your comments:
The characteristics of the GPR used should be better explained.
We primarily utilize low frequencies in our GPR system, ensuring that the frequencies used for other components on the drone are entirely different. This deliberate frequency selection prevents any interference between the GPR and other drone equipment, guaranteeing optimal performance and accurate data collection.
Not only can impediments and obstacles at the crime scene create anomalies or false positives, but the presence of forensic evidence or trace evidence on the surface can also be compromised by flying too low due to the drone blades creating a wind effect.
We appreciate your insight regarding the potential impact of flying the drone too low on the surface. In response, we have adjusted the relevant section after discussing flight altitudes. We have clarified that we flew the drone at an optimal height to ensure clear visibility of the graves and these preliminary results did not take into account possible surface contamination. However, we acknowledge the need for future research to explore different altitudes and investigate the possibility of data loss at higher flight altitudes. Different crime scenes may require varying flight altitudes, and it is crucial to examine how we can effectively address these requirements. For example, crime scenes with a significant amount of surface traces may require higher flight altitudes, but we must also consider potential data loss at higher elevations.
The issue of using pigs as human body mimics is a highly debated topic, and it is important to explore this aspect further, especially from a crime scene perspective.
We acknowledge the ongoing debate surrounding the use of pigs as human body analogues. In response, we have added a section to the discussion specifically addressing this concern. We recognize that while pigs may not be an ideal substitute for human remains, they were suitable for obtaining preliminary results in our study. However, we emphasize the necessity of transitioning to the use of human remains in future research. This addition acknowledges the limitations of using animal models and underscores the importance of working with human remains in order to provide more accurate and applicable findings for crime scene investigations. We have also included references to relevant studies that contribute to the debate and encourage further exploration of this topic.
The introduction should also consider other drone+GPR approaches, such as the "traditional" approach of evaluating data collected by drones and analyzing GPR data from the ground.
We have revised the introduction to incorporate a discussion of other drone and GPR approaches, including the "traditional" approach you mentioned. We have highlighted that despite the different methodologies, the resulting GPR data remains consistent, as it can be converted from 2D to 3D representations. Additionally, we have acknowledged previous studies that have utilized drones to support GPR research, particularly through the use of surveying drones. These additions provide a broader context for our work and demonstrate the existing body of research in this field.
Once again, we sincerely appreciate your thoughtful feedback, which has significantly contributed to improving the clarity and completeness of our paper. The revisions we have made in response to your suggestions have strengthened the overall quality and enriched the content of our research. We are confident that these changes address the issues you raised and further enhance the contribution of our study.
Thank you for your time and expertise in reviewing our paper.
Sincerely,
Louise Lijcklama a Nijeholt
Reviewer 3 Report
This is an excellent paper, and I applaud your team for an interesting study. However, I would like to suggest inclusions (that should be considered minor) but without which your paper's impact is severely limited. I work in forensic/ archaeology/GPR, and at present after reading your paper, I am not going to go out and buy a drone-based GPR as I am still not convinced it "works" (can accurately produce grave responses). However, you have a very clear and well-reasoned research study at the Benelo Facility, just underexplained currently in this piece. If this paper is expanded with suggested revisions, your paper could be much more significant.
1. Greater detail on your results is needed. You have an excellent methodological study but currently, the description of the results leaves the reader with a lot of questions. First, how did the GPR results compare in depth and response to the known depths/ variables of the simulated graves? A table showing your known values when creating the graves and those generated by the drone-based GPR system would greatly improve your research impact. Second, further description about what kind of geophysical responses you see in the radargrams for the graves is necessary (otherwise we have to just believe that what you are saying is correct). For instance, it would be good to consult archaeology/forensic references that dissect radargrams for burial traits/language so that you can add descriptions for what you see in each of the simulated graves. Greater description of each of the graves would also be an asset. Why don't you show any depth slices for the graves?
2. Greater description and comparison to ground-based GPR of unmarked graves results is needed. If you did not take ground-based measurements, that is alright, but a discussion of how your results compare to these would benefit this paper. Specifically, how close can you get your flight lines compared to ground-based GPR transects? "High resolution" ground-based GPR grids are less than 25 cm transect spacing. As a terrestrial GPR person, we do not identify graves just by hitting the target once, but only when we consistently capture a target over many lines spaced 25 cm or less (e.g., 4+ GPR lines showing the same hyperbolic feature is needed to identify a grave). So further discussion about the current limitations of the technique, future directions, and how it could be employed in its current state would also benefit this paper.
3. To this end, added discussion and reference to current best practices in ground-based GPR is essential. Situating this paper within the broader unmarked graves work done in archaeology/forensic GPR and what contributions this new sensor-system could make is essential. Currently, the reference list is stunted.
4. Please decide on a clear and consistent GPR terminology (and cite your source). You use B-scan as well as radargram. Although I know what you are saying, GPR scholars are very bad at adding new terminology to describe the same old thing. A little bit of description of how you use the terms would go a long way.
5. What do you mean by "control graves" do these have human remains in them or are these empty?
6. Providing a definition for clandestine grave (or other types of burials) would improve this contribution as well.
Primarily minor spelling and grammatical edits throughout the document are needed.
Some very noticeable phrasing that should be changed or clarified:
1. "Geophysical sensing techniques" should just be "geophysical techniques"
2. Last paragraph of section 1, particularly lines 122-126 should be clarified as I do not immediately understand what you are trying to say.
3. Line 178, please define what you mean by "platforms" or change the word. You use platforms to describe very different things.
Author Response
Dear reviewer,
We would like to express our gratitude for taking the time to review our article titled Utilizing drone-based GPR for crime investigations in localizing and identifying clandestine graves. Your feedback and suggestions have been invaluable in improving the clarity and impact of our research. In response to your comments, we have made the following revisions:
Greater detail on results:
We have now included a table that presents the known depths/variables of the simulated graves alongside the depths generated by our drone-based GPR system. Furthermore, we have consulted archaeology and forensic references to describe the geophysical responses observed in the radargrams for each of the simulated graves. Additionally, we have provided greater descriptions for each of the graves, offering more insights into the characteristics and features observed. While we have not included depth slices for the graves, we acknowledge the importance of this information and will consider its inclusion in future work.
Greater description and comparison to ground-based GPR of unmarked graves:
We apologize for the oversight in not providing a discussion or comparison to ground-based GPR results. We acknowledge the significance of this aspect and will rectify it in the revised manuscript. We will specifically address the proximity of our flight lines to ground-based GPR transects, considering that "high resolution" ground-based GPR grids typically have less than 25 cm transect spacing. We understand that consistent detection of a target over multiple lines is essential in identifying graves. We will discuss the current limitations of our technique, outline future directions, and highlight the potential application of our system in its current state. Moreover, we appreciate your input on the technical feasibility of our proposed approach, and we will address the challenges related to planning runways and the accuracy of drone flight lines.
Inclusion of current best practices and references:
We acknowledge the need to situate our paper within the broader context of unmarked graves work in archaeology/forensic GPR. We have included additional discussion and references to current best practices in ground-based GPR. By doing so, we aim to emphasize the contributions our new sensor system can make to the field. We apologize for the limited reference list in the previous version and have expanded it to encompass relevant literature and studies.
Clear and consistent GPR terminology:
We appreciate your suggestion regarding the GPR terminology used in the article. To address this concern, we have revised our terminology throughout the manuscript to consistently use the term "radargram" instead of "B-scan." We apologize for any confusion caused by the inconsistency and appreciate your input on improving the clarity of our terminology.
Definition of "control graves":
To clarify any ambiguity, we have added a definition of "control graves" in the paper. We specify that control graves in our study refer to graves without any cadavers, serving as a comparison for the graves containing cadavers.
Definition of clandestine grave:
We have incorporated a definition of clandestine graves within the article. This addition will enhance the contribution of our research by providing a clear understanding of the terminology used in the context of the study.
Comments on the quality of English language:
We have carefully reviewed and made the necessary spelling and grammatical edits throughout the document. We have also addressed the phrasing concerns you raised, ensuring clarity and coherence. Specifically, we have modified the term "geophysical sensing techniques" to "geophysical techniques" and provided clarification in the different sections.
Once again, we would like to express our gratitude for your valuable feedback, which has significantly enhanced the quality of our paper. We believe that the revisions we have made in response to your suggestions have strengthened the paper and improved its readability. We are confident that these changes will address the concerns you raised and provide a more comprehensive understanding of our research.
Thank you for your time and consideration. We look forward to hearing from you soon.
Sincerely,
Louise Lijcklama a Nijeholt
Round 2
Reviewer 2 Report
Dear Authors,
I have appreciated the improvements you did. But since you have mentioned the ground-based gpr methodology, you should provide also a practical comparison with real data acquired with the same gpr system. Otherwise, this comparison is partial and not providing any relevant information. Moreover, please improve conclusions accordingly.
Author Response
Dear Reviewer,
Thank you for your feedback on our research. We appreciate your recognition of the improvements we made. We would like to address your concern regarding the lack of a comparison with data acquired using the ground-based GPR system.
We apologize for any confusion caused by the omission of this comparison in our study. The primary focus of our research was to explore the integration of GPR with a drone-based platform, rather than directly comparing ground-based GPR to drone-based GPR. We acknowledge the importance of such a comparison to validate our findings, and we understand that it would provide relevant information to the field.
In light of your feedback, we agree that a comprehensive comparison between ground-based GPR and drone-based GPR should be conducted in future research to further verify our findings. We plan to extend our research and include this comparative analysis in our future studies. We have now explicitly mentioned in our paper that we did not perform this comparison in the current research, and we hope this clarification makes it clear to the readers.
Furthermore, we will also revise the conclusions of our paper to reflect the limitations of the current study and emphasize the need for further research that includes a practical comparison of ground-based GPR with real data.
Once again, we appreciate your valuable feedback and the opportunity to address this issue. Please let us know if you have any further suggestions or concerns. We remain committed to improving the quality and relevance of our research.
Sincerely,
Louise Lijcklama a Nijeholt
Reviewer 3 Report
I am very happy with the changes you made. This is a great article that should be published. There are still a few typos/grammar issues in your "track changes" document that I recommend you take another look at it. Otherwise, great job!
Some typos and grammar issues detected.
Author Response
Dear Reviewer,
Thank you for your positive feedback and your satisfaction with the changes we made to the article. We greatly appreciate your kind words and your recommendation for publication.
We acknowledge your comment regarding the remaining typos and grammar issues in the "track changes" document. We apologize for any oversights on our part. We have thoroughly reviewed the document once again to address these concerns.
We are grateful for your keen eye and attention to detail, as it contributes to the overall quality of our work. Your feedback is invaluable to us, and we want to ensure that the article is polished and error-free.
Thank you again for your positive review and for bringing the remaining issues to our attention. We took immediate action to rectify them and deliver a final manuscript that meets the highest standards.
If you have any further suggestions or comments, please feel free to let us know. We are committed to continuously improving our work and appreciate your support throughout this process.
Sincerely,
Louise Lijcklama a Nijeholt